# The Effectiveness of Parent Training Programs for Child Maltreatment and Their Components: A Meta-Analysis

**DOI:** 10.3390/ijerph16132404

**Published:** 2019-07-06

**Authors:** Jeanne Gubbels, Claudia E. van der Put, Mark Assink

**Affiliations:** Research Institute of Child Development and Education, University of Amsterdam, Nieuwe Achtergracht 127, 1018 WS Amsterdam, The Netherlands

**Keywords:** child maltreatment, child abuse, parent training program, effectiveness, program components, meta-analysis

## Abstract

This is the first meta-analytic review investigating what components and techniques of parent training programs for preventing or reducing child maltreatment are associated with program effectiveness. A literature search yielded 51 studies (*N* = 6670) examining the effectiveness of parent training programs for preventing or reducing child maltreatment. From these studies, 185 effect sizes were extracted and more than 40 program components and techniques were coded. A significant and small overall effect size was found (*d* = 0.416, 95% CI (0.334, 0.498), *p* < 0.001). No significant moderating effects were found for contextual factors and structural elements (i.e., program duration, delivery location, and delivery setting). Further, no significant moderating effects were found for most of the coded program components and techniques, indicating that these components are about equally effective. Only a few program components and techniques moderated program effectiveness, however these effects were negative. These results indicated that improving parental personal skills, improving problem solving skills, and stimulating children’s prosocial behavior should not be the main focus of parental training programs for preventing and reducing child maltreatment. This also holds for practicing new skills by rehearsal and giving direct feedback in program sessions. Further clinical implications and directions for future research are discussed.

## 1. Introduction

Child maltreatment is a major problem that affects many children around the world and has serious consequences for individual victims and society. A review of a series of meta-analyses on the prevalence of child maltreatment showed that prevalences varied from 12.7% to 36.3% for self-report studies and from 0.3% to 0.4% for multi-informant studies [1]. Child maltreatment contributes substantially to child mortality and morbidity, and has long lasting negative effects, such as physical, behavioral, and psychological problems [2,3,4]. Given the great individual and societal impact of child maltreatment, it is crucial to prevent child maltreatment. Therefore, many interventions aimed at preventing or reducing child maltreatment have been developed throughout the years. Although various meta-analyses on the effectiveness of these interventions showed limited overall effects [5,6,7,8], the meta-analysis of Van der Put et al [9] showed that parent training programs are one of the most effective programs in preventing child maltreatment, compared with other types of programs, such as home visitation programs. However, why that is, remained to be explored. Therefore, the present study aimed to examine what components and techniques contribute to this relatively strong effect of parent training programs. Examining this is important, as effectiveness of parent training programs may be increased by adding components that are positively associated with effectiveness and leaving out components that are negatively associated with effectiveness. In this way, it is to be expected that child maltreatment rates can be further reduced.

Worldwide, many programs have been designed to prevent child maltreatment. These programs are usually based on etiological models for child abuse and neglect. Generally, it is assumed that child maltreatment is caused by a complex interaction between multiple risk factors, rather than the presence of only a single risk factor [10,11]. Belsky [10,11] drew on the ecological model of Bronfenbrenner [12,13] and posed that the risk of child maltreatment is determined by the interaction of risk factors at four different levels, which are: (1) the history of parents/caregivers who abuse their child (ontogenetic development of parents); (2) characteristics of the child and the family (microsystem); (3) characteristics of the community in which the family lives and the degree of social support in the social environment surrounding the family (exosystem); and (4) the attitude of the society towards children and maltreatment (macrosystem). Belsky assumed that risk factors more proximal to the child (levels one and two) exert more influence than risk factors in more distal social systems (levels three and four). Further, the personal psychological resources of parents are considered to be central factors as they mediate more distal influences [14]. When looking at the results of some review studies on risk factors for child abuse and neglect, parent and family related risk factors are considered most important [15,16,17]. Therefore, parents are often the target of programs designed to prevent or reduce child maltreatment. 

An important question is how parent programs must be shaped in such a way, so that child maltreatment is prevented or reduced effectively. Currently available parent training programs with this aim generally focus on improving child-rearing skills and parenting practices, and on modifying parental attitudes towards harsh parenting. In many of these programs, individual or group-based parenting support is offered. Components and techniques that are commonly found in these programs are: stimulating a positive parent-child interaction, increasing parental knowledge of child development, improving discipline strategies and behavior management, and improving parental attitudes towards their child and/or parenting in general. Most programs also comprise components aimed at enhancing the emotional well-being of parents, for instance by learning parents how to control their anger and stress [18]. 

Most parent training programs are targeted at a clearly defined population. Programs aimed at preventing the occurrence of child maltreatment in at-risk, but non-maltreating families can be distinguished from programs aimed at reducing (episodes of) child maltreatment in maltreating families. The ACT-Parents Raising Safe Kids program (ACT-PRSK) [19] is an example of a program aimed at preventing family violence and child maltreatment, which is available to all parents of young children regardless of the risk for child maltreatment in these families. In this program, groups of parents and caregivers are trained in effective parenting, by learning them for instance how to discipline their child in a nonviolent way, how to control their anger, how to solve social problems, and what a “normal” child development looks like. In their randomized controlled trial (RCT), Portwood et al. [20] found a reduction in harsh verbal and physical disciplining in the ACT-PRSK group compared with the comparison group. Moreover, Knox et al. [21] found lower rates of child-oriented psychologically and physically aggressive behavior of parents who received ACT-PRSK in community health centers. A well-known parent training program that aims to *reduce* the incidence of child maltreatment in physically abusive parents is Parent–child Interaction Therapy (PCIT). PCIT is a protocolled program focusing on the interaction between the parent and the child as well as the child’s behavior [22]. In live-coached sessions, parents are trained in skills regarding child-directed interaction, such as giving positive attention and praising good behavior, and in skills regarding parent-directed interaction, such as behavior management strategies like setting rules, responding consistently, and using the ‘time-out’ technique. A recent meta-analysis on the effects of PCIT showed a significant reduction in parental stress and externalizing behavior of children [23]. A different RCT study further showed that parents assigned to PCIT had fewer re-reports for physical abuse compared with parents who were assigned to standard community services [24].

A number of meta-analyses have synthesized results on the degree to which parenting training programs reduce (the parents’ risk of) child maltreatment [5,7,18,25,26]. These reviews generally found (very) small to moderate overall effects, according to Cohen’s [27] criteria for interpreting effect sizes. More specifically, Euser et al. [5] found a significant, but only very small effect (*d* = 0.13) of parent training programs aimed at preventing or reducing child maltreatment, which even became non-significant after controlling for publication bias. Pinquart and Teubert [26] found the same effect size for parent education interventions offered to expectant and new parents. Chen and Chan [25] found a significant and slightly higher (but still modest) effect (*d* = 0.30) of parent training programs on substantiated and self-reported child maltreatment reports and child maltreatment potential. Finally, Lundahl et al. [18] found a significant and moderate effect (*d* = 0.45) of parent training programs on child maltreatment. However, these authors only included three studies that reported on actual child maltreatment. In only a few of these meta-analyses, potential moderators of program effects were examined. It was found that the effectiveness of parent training programs could vary by several study and program characteristics, like sample type (i.e., at-risk samples vs. maltreating samples) [5,26,28], the delivery setting of the program (individual vs. group, in-home vs. office setting) [18], and the length of the program [5]. 

To better understand why parent training programs are effective, it is important to examine how specific intervention components, such as different types of program content, techniques, or strategies, influence intervention effectiveness. In previous literature, various terms were used for these intervention components. Chorpita et al. [29] used the term *practice elements* for discrete clinical techniques or strategies used as part of a larger intervention plan (e.g., “relaxation”, “exposure” or “psychoeducation”). They used the term *common elements* for practice elements that are commonly found in different effective treatments. Another term for intervention components is active ingredients. Barth and Liggett-Creel [30] (pp. 7) defined an active ingredient as an “element of treatment, which has been found to make a reliable positive difference”. Finally, Blase and Fixsen ([31], pp. 3) label these intervention components as *core components*, which they define as “the essential functions or principles, and associated elements and intervention activities, that are judged necessary to produce desired outcomes”. They argue that these core components refer to program characteristics, like contextual aspects (e.g., the delivery setting and the location of the sessions), structural elements (e.g., the duration of the program and intensity of the sessions), and specific intervention practices (e.g., teaching problem-solving skills to parents, improving parental communication skills, practicing social skills with parents, and reinforcing appropriate parental behavior). 

Van der Put et al. [9] investigated in their meta-analysis which components are positively associated with effectiveness of interventions for preventing or reducing child maltreatment. They found larger effect sizes for interventions with (one of) the following three components: increasing self-confidence of parents (for preventive interventions), improving parenting skills, and providing social and/or emotional support (both for curative interventions). Whereas Van der Put et al. [9] examined the program components of all types of interventions in their meta-analysis, Temcheff et al. [32] recently reported on common components of evidence-based parenting programs for preventing maltreatment of school-age children. In their review they identified the components that were common to these programs. They found that most evidence-based programs included components such as improving the parent-child communication, regulating emotions of parents, improving parenting skills, and providing parent education. However, with the method that Temcheff et al. [32] used, they could not determine to what extent specific program components contribute to the effectiveness of parenting programs. This was done by Kaminski et al. [33], who examined which components were associated with the effect of parent training programs on parenting behaviors and children’s externalizing behavior. They found that components such as stimulating positive parent-child interactions, improving emotional communication skills (e.g., active listening), and teaching parents the use of time-out were associated with larger program effects. However, a meta-analytic review on the effectiveness of program components of parent training programs in preventing or reducing child maltreatment was not yet available. Therefore, this review aimed at examining the moderating effect of different program components on the overall effectiveness of parent training programs by conducting a three-level meta-analysis. Specifically, we examined the moderating effects of contextual factors (i.e., delivery setting, the program’s aim), structural elements (i.e., the program’s duration, the average number of sessions, and the interval between sessions), specific intervention practices (i.e., improving the parent-child communication, improving parental supervision, setting clear rules and consequences, positive reinforcement), and delivery techniques (i.e., modelling, role-playing, video-feedback).

## 2. Materials and Methods 

### 2.1. Inclusion Criteria

Studies were selected if they met the following three inclusion criteria. First, studies had to report on the effect of at least one parent training program for preventing or reducing child maltreatment. These programs had to be aimed at improving parenting skills, either in an individual or group setting. However, home visitation programs, which include prenatal and early-childhood home visits as preventive services to families in need of support [34], were not included. These programs are offered as a first step in prevention, and delivered to pregnant women or parents of very young children. Therefore, the components of these programs could fundamentally differ from the components of the parent training programs that we aimed to examine in this meta-analysis. As for the concept of child maltreatment, this was defined as any acts of commission or omission by a parent or other caregiver that result in harm, potential for harm, or threat of harm to a child [35]. Consequently, we included studies that reported on physical abuse, sexual abuse, and different forms of neglect. In addition, studies examining the effect of programs on child abuse potential, harsh parenting (such as corporal/physical punishment or parental aggression toward children), and out-of-home placement were also included. Second, the design of a primary study had to include a control group, which could consist of a ‘no care’ control group, a waiting list control group, a ‘service as usual’ control group, or a control group receiving minimal care (one session) or written materials. Both randomized controlled trials (in which participants were randomly assigned to either the intervention or the control group) and quasi-experimental studies (no random assignment) were included. Third, studies had to report on at least one effect size or sufficient information to calculate an effect size.

### 2.2. Selection of Studies

To select relevant studies for this meta-analysis, multiple searches were conducted. First, we screened the full reference list of Van der Put et al. [9], who aimed their meta-analytic review at identifying effective components of child maltreatment interventions. The authors analyzed studies on the effect of several types of interventions on child maltreatment outcomes, including studies on the effects of parent training programs. Second, the electronic databases Google Scholar, Web of Science, PsycINFO, Sciencedirect, and Educational Resources Information Center (ERIC) were searched for relevant articles, reports, dissertations, books, and chapters published before July 2018. Studies were searched using the following keywords regarding (a type of) child maltreatment, intervention features, study design, and caretakers in different combinations: ‘child *’, ‘abus *’, ‘maltreat *’, ‘neglect *’, ‘interven *’, ‘prevent *’, ‘program *’, ‘training’, ‘randomized’, ‘evaluat *’, ‘assess *’, ‘experiment *’, ‘parent *’, and ‘caregiv *’. In additional searches, these keywords were combined with the full names and abbreviations of several well-known parent training programs, such as Triple P, Parent-Child Interaction Therapy (PCIT), and Incredible Years (IY). Finally, the references of relevant review studies [5,7,18,25,26,33,36,37,38,39] were examined to search for additional studies that may have been missed in the electronic search. The searches resulted in 3713 studies. After removing duplicates, 925 studies were screened based on their title and abstract. In the screening phase, 676 studies were excluded because of their irrelevance to the subject of this meta-analysis (e.g., studies examining other types of programs or other outcomes). Of the remaining 249 relevant studies the full text was evaluated. Finally, 51 studies met all inclusion criteria and were included in the current study. A flowchart of the search procedure is presented in Figure 1 and Table A1 shows several characteristics of the included studies.

### 2.3. Coding the Studies

A detailed coding scheme was designed according to the guidelines of Lipsey and Wilson [40] to code relevant study and program characteristics that could be tested as moderators of the overall effect of parent training programs. Regarding study characteristics, we coded information on the publication year, sample type (general sample, risk group, maltreating sample), sample size, age of the children, parental age, percentage of cultural minorities in samples, research design (RCT, quasi-experimental with matching, quasi-experimental without matching), intent-to-treat design (yes/no), a follow-up of at least 12 months (yes/no), study quality index (a numerical score based on the three previous described variables), type of control group (treatment-as-usual, no treatment, waiting list, other), type of outcome measure (official report, parent-report, child-report, observation) and the length of the follow-up duration in months. 

The program characteristics were divided into contextual factors, structural elements, program components, and delivery techniques. As for the contextual factors, we coded the general aim of the study (reduction of (re-)abuse or prevention), delivery location (home/ambulant, treatment center, by telephone, online, other), and delivery setting (only with parents, both parents and child, parent group without children). The coded structural elements were the program duration (0–12 weeks, 13–24 weeks, >24 weeks), minimum and maximum program duration (in weeks), average number of program sessions (attended by the participants), and the interval between program sessions (multiple sessions a week, weekly, every other week/monthly, ascending/descending intensity). 

The coded program components and delivery techniques are described in Table 1, and were coded as present or absent depending on whether the program applied the component/technique. In order to determine this for each component and technique, we not only carefully read information about a program in a primary study, but we also read factsheets, manuals, or protocols on a parent training program, which were written by program developers and could most often be found online.

In the first coding round, the first and the last author of this study independently coded five randomly selected studies that were eligible for inclusion (reporting on 24 effect sizes in total). These independent codings were compared, and percentages of agreement were calculated. For the study characteristics, the contextual factors, and the structural elements, the interrater agreement was 85%. The agreement was 80% for the double-coded program components and delivery techniques, and 75% for the double-coded effect sizes. All inconsistencies in the independent codings were discussed and resolved until the authors fully agreed on all final coding decisions. In general, minor coding errors caused the disagreement in the double-coded study characteristics, contextual factors, structural elements, and effect sizes. The discrepancies in the coded program components and techniques were mostly due to different coding styles. Some codings could be based on rather elaborate information about the parent training programs as described in the studies or factsheets, whereas other codings could only be based on a rather narrow definition of a program. Prior to coding the remaining studies, the coding sheet was modified where necessary. In a second coding round, it was decided to code the remaining 46 studies according to a more strictly coding style to reduce subjectivity in coding as much as possible. This coding round was performed by the first author of this study. Whenever the first author doubted about the presence of a certain component or technique, the other two authors were consulted.

### 2.4. Calculation of Effect Sizes

The outcomes of the primary studies were transformed into the standardized difference between two means, also referred to as Cohen’s *d*. Most studies reported on means and standard deviations, proportions, and odd ratios. These outcomes were transformed into Cohen’s *d* values using formulas of Ferguson [41], Lipsey and Wilson [40], and Rosenthal [42]. As for the direction of effect sizes, a positive *d* value indicated that lower levels of child maltreatment (or other assessed outcomes, such as child abuse potential, harsh parenting, or out-of-home placement) were found in the intervention group than in the control group, whereas a negative *d* value indicated that higher levels of child maltreatment were found in the intervention group than in the control group. If results in primary studies were described as non-significant without any statistical information, a *d* value of zero was coded [43]. This procedure was applied to one study reporting on one effect size. 

All coded variables and calculated effect sizes were entered in SPSS version 24 (SPSS Inc., Chicago, IL, USA). Next, continuous variables were centered on their mean, and categorical variables were recoded into dummy variables.

### 2.5. Statistical Analyses

To estimate the overall effect of parent training programs on child maltreatment and to examine which study or program characteristics moderated this effect, a three-level meta-analysis was conducted. Because most studies reported on multiple relevant effect sizes, a traditional random effects model was extended to a three-level random effects model so that effect size dependency is accounted for [44]. As a result, there is no need for aggregating or selecting data, implying that all relevant information can be extracted from primary studies and maximum statistical power can be achieved [45]. In our meta-analytic model, three forms of effect size variation were taken into account: the random sampling variation of observed effect sizes (level 1), the variance between outcomes within studies (level 2), and the between-study variance (level 3) [44,45,46,47,48]. In estimating the overall effect, effect sizes from primary studies were weighted by the inverse of their variance (i.e., sampling error), so that effect sizes derived from larger studies contributed more to the overall effect size estimate than effect sizes derived from smaller studies. Next, to determine whether significant variance was present at level 2 or 3 of the model, two likelihood ratio tests were performed. In these tests, the deviance of the full model was compared to the deviance of a model excluding the variance parameters of either level 2 or 3. In case of significant variance on level 2 and/or 3, the distribution of effect sizes was considered to be heterogeneous. This indicates that the effect sizes could not be treated as an estimate of a common effect size, and thus, moderator analyses were performed to test variables that may explain variance in effect sizes. The program R (version 3.5.0) (R Foundation for Statistical Computing, Vienna, Austria) and the metafor-package [49] were used to build the 3-level meta-analytic models. The model was extended by including study and program characteristics as covariates, so that their influence on the overall effect of parent training programs on child maltreatment could be examined. We used the R syntax as described by Assink and Wibbelink [45]. In all analyses, a 5% significant level was used.

### 2.6. Bias Assessment

A common problem in conducting a meta-analysis is that studies with non-significant or negative results are less likely to be published than studies with positive and significant results. The effect sizes extracted from the primary studies included in the current meta-analysis may therefore not be an adequate representation of the actual effect of parent training programs on child maltreatment. This phenomenon is called publication bias and is often referred to as the ‘file drawer problem’ [50]. Further, the results of a meta-analysis could be affected by other forms of bias, such as coding or selection bias. In order to examine the degree to which our results were affected by (different forms of) bias, we conducted a nonparametric and funnel plot-based trim-and-fill analysis as described by Duval and Tweedie [51,52]. A funnel plot is a scatter plot of the effect sizes against the effect size’s precision (1 divided by the standard error). In this analysis, the symmetry of the funnel is tested. In case of publication bias, a “gap” in the effect size distribution is present, leading to asymmetry of the funnel plot. This asymmetry is restored by imputing “missing” effect sizes that are calculated on the basis of existing effect sizes in the data set. Subsequently, a “corrected” overall effect can be estimated in a sensitivity analysis using the data set to which the imputed effect sizes that were produced by the trim-and-fill algorithm have been added. In this way, the degree to which the results were affected by bias can be determined. The trim-and-fill analysis was conducted using the “trimfill” function of the “metafor” package [49] in the program R (Version 3.5.0).

As one of the included studies provided 56 effect sizes [53], which was about 30% of the total amount of effect sizes in this meta-analysis, a sensitivity analysis was performed. With this analysis, we could test whether the inclusion of this study changed the overall effect. Therefore, an overall effect based on the dataset without Kolko [53] was compared with the overall effect based on the full dataset.

## 3. Results

The current meta-analysis consisted of 51 studies (with *k* = 50 non-overlapping samples), reporting on 185 effect sizes and a total of *N* = 6670 participants, of whom *n* = 3340 participated in a parent training program and *n* = 3330 participated in a control group. The sample sizes of the included studies varied between *n* = 18 and *n* = 918. The studies were published between 1985 and 2018, and were conducted in the USA (*k* = 26), Europe (*k* = 8), Canada (*k* = 3), Australia or New Zealand (*k* = 6), and in various other non-western countries (*k* = 7).

### 3.1. Overall Effect

Table 2 presents the results for the overall effect of parent training programs on child maltreatment. A significant overall effect was found with a Cohen’s *d* of 0.416; 95% CI (0.334, 0.498), *t* (184) = 9.977, *p* < 0.001. According to the guidelines formulated by Cohen [27] to interpret the magnitude of effect sizes, with effect sizes of *d* = 0.20 considered small, *d* = 0.50 medium, and *d* = 0.80 large, this effect is small. The results of the two log-likelihood ratio tests showed that significant variance was present both at level 2 (*χ^2^* (1) = 17.611, *p* < 0.001; one-sided) and level 3 (*χ^2^* (1) = 3.712, *p* = 0.027; one-sided) of the meta-analytic model.

Of the total variance, 33.2%, 39.3% and 27.4% was distributed at levels 1, 2 and 3, respectively. As these results indicated substantial heterogeneity in effect sizes, we could test study design and intervention characteristics as potential moderators of the overall effect of the parent training programsNext, we performed a sensitivity analysis, as the study of Kolko [53] produced a substantial number of effect sizes (i.e., 56 effect size could be extracted; see also the Method section). This resulted in an overall effect of 0.425; 95% CI (0.331, 0.518), *t* (128) = 8.976, *p* < 0.001 (see Table 2). This overall effect did not substantially differ from our initial estimated overall effect (Δ*d* = 0.009). Therefore, it could be assumed that the effect sizes reported in study of Kolko [53] did not substantially affect the overall effect. 

Finally, the trim and fill analysis showed that the distribution of effect sizes was symmetrical (see the funnel plot in Figure 2), implying that bias was not present in the data that were synthesized, and that imputation of effect sizes was not necessary. Therefore, the overall effect was not re-estimated.

### 3.2. Moderator Analyses

Each potential moderator of interest was examined in a bivariate model. The results of these analyses can be found in Table 3, in which potential moderators are classified into study design characteristics, contextual factors, structural elements, program components, and delivery techniques.

#### 3.2.1. Study Design Characteristics

As for the study design characteristics, we found a significant moderating effect of the study’s sample size (larger sample sizes yielded smaller effect sizes). Further, a significant moderating effect of the type of research design was found. Larger effect sizes were found in studies with a quasi-experimental design, in which no matching was used to assign participants to either the intervention or control group (*d* = 0.805), in comparison with RCTs (*d* = 0.358). A significant moderating effect was also found for the study quality index, indicating that effect sizes decreased as the quality of studies increased.

#### 3.2.2. Contextual Factors

None of the coded contextual factors, including the general aim of the program, the delivery location, and the delivery setting, significantly moderated the overall effect of parent training programs.

#### 3.2.3. Structural Elements

None of the structural elements, including the duration, average number of sessions, and the interval of the sessions, significantly moderated the overall effect of parent training programs.

#### 3.2.4. Program Components

Several program components significantly moderated the overall effect. Notably, the presence of these components reduced program effects. In specific, smaller effect sizes were found for programs with a focus on improving personal skills of parents (*d* = 0.373 versus *d* = 0.816 for programs without this component), improving parental problem solving (*d* = 0.363 versus *d* = 0.512 for programs without this component), and stimulating prosocial behavior/discourage antisocial behavior of children (*d* = 0.361 versus *d* = 0.527 for programs without this component).

#### 3.2.5. Delivery Techniques

As for the delivery techniques, we also found a negative moderating effect. Specifically smaller effects were found for programs using practice and rehearsal as a delivery technique (*d* = 0.329 versus *d* = 0.512 for programs not using this technique).

## 4. Discussion

The aim of the current meta-analysis was to gain insight into the program components that are associated with the effectiveness of parent training programs for preventing or reducing child maltreatment. To meet this aim, we examined the moderating effect of multiple program components on the overall effect of parent training programs. First, we found a significant overall effect of *d* = 0.416 of all included parent training programs on child maltreatment, and the results of the trim-and-fill analysis indicated that no bias was present in the data. Similar effect sizes were found in previous meta-analyses examining the overall effect of parent training programs on child maltreatment [9,18] or on parenting behavior [28,33].

### 4.1. Study Characteristics

Our findings indicate that studies with smaller sample sizes yielded larger effect sizes. This finding was in line with the pattern described by Sterne et al. [54], who found that studies with small sample sizes are more likely to report beneficial effects of interventions than larger studies. This so-called “small-study effect” may arise from (a combination of) publication bias, bias forms caused by a lower methodological quality of small studies [55], and true differences in the underlying effects between smaller and larger studies. As the results of the trim-and-fill analysis suggested that no bias was present in the data, it is not likely that a publication bias caused such an effect. However, there are several methodological shortcomings of the trim-and-fill method that should be taken into account here [56,57,58]. 

Further, larger effect sizes were found in studies with quasi-experimental designs using non-matched intervention and control groups relative to RCTs. This was in line with previous literature, as meta-analyses on the effect of interventions on child maltreatment outcomes that included only RCTs generally showed smaller effect sizes than meta-analyses including both RCTs and quasi-experimental designs. For example, Euser et al. [5] found a very small effect (*d* = 0.13) of RCTs on interventions for preventing or reducing child maltreatment. Pinquart and Teubert [26] found a similar effect (*d* = 0.13) of RCTs on parenting interventions for families with newborns. Generally, the effect of interventions can best be determined in RCTs, as random assignment of participants to an experimental and a control group (theoretically) equalizes both groups on all other variables. Therefore, RCT’s are considered to be the most powerful study design in intervention research [59,60]. On the other hand, Van der Put et al. [9] noted that RCTs are rare in the field of child maltreatment, and that consequently, essential information would be missing in a review of only primary studies with an RCT design. 

Finally, we found that studies with a lower study quality index showed larger effects than high-quality studies. This index was determined for each study and was based on three aspects of a primary study’s design: the research design of the study (RCT vs. quasi-experimental), whether or not the study used an ‘intent-to-treat’ design, and whether or not the study included a 12 month (or longer) follow-up assessment of the child maltreatment outcomes. Out of these three variables, the research design was the only variable with a significant moderating effect, indicating that the significant moderating effect of the study quality index was largely explained by this variable.

### 4.2. Contextual Factors, Structural Elements, Program Components, and Techniques

Our findings indicate that the contextual factors and structural elements that we investigated, were not associated with the effectiveness of parent training programs for preventing or reducing child maltreatment. Therefore, it is not expected that programs that differ on these characteristics will also differ in their effectiveness. This is not in line with the findings of Van der Put and colleagues [9] who found moderating effects for several structural element of child maltreatment interventions. However, they focused on a wide variety of child maltreatment interventions, including multisystemic interventions and home visiting programs, which are all very different in nature. The structural elements and contextual factors of the included parent training programs in this meta-analysis are more similar, which may explain why these variables didn’t significantly moderate the overall effect. Furthermore, our findings might be explained by the discrepancy between information that was used to code these characteristics (derived from factsheets, protocols and manuals) and the way the programs were actually performed within the research groups of the included studies. Possibly, some of the programs were performed more flexible instead of strictly according to the protocol. This makes it is difficult to categorize these programs with regard to their contextual factors and structural elements, as in reality these characteristics might be different. This should be taken into account when interpreting these results. However, there might be other structural and contextual factors, not investigated in the current study, that are related to the effectiveness of parent training programs.

Furthermore, we found no significant moderating effects for most of the program components and delivery techniques. This suggests that the different program components and techniques used in parent training programs are about equally effective in preventing or reducing child maltreatment. Previous meta-analyses did find moderating effects of program components and techniques for other types of interventions aimed at preventing or reducing child maltreatment, such as family programs, or substance abuse interventions [9,34]. Generally, previous literature suggests that identifying effective components of interventions is important to understand why interventions are effective (or ineffective) in achieving a certain outcome for a certain person or family [29]. In addition, this knowledge is useful for improving interventions by integrating effective components in interventions and/or eliminating ineffective components from interventions. However, the results of the current meta-analysis indicate that the effectiveness of parent training programs for the prevention or reduction of child maltreatment cannot be improved by adding or leaving out the components that were currently examined, as these components are about equally effective.

For three program components and one program technique, a negative moderating effect was found. Parent training programs targeting the personal and problem-solving skills of parents showed significant smaller effects on child maltreatment compared with programs not targeting these skills. Several review studies showed that personal problems of parents, such as stress, anger and health problems, are important risk factors for child maltreatment [15,16,17]. Therefore, it could have been expected that addressing these problems in parent training programs contributes to the prevention or reduction of child maltreatment. However, our findings are in line with those of Kaminski et al. [33], who examined the components associated with the effect of parent training programs on positive parenting behaviors and children’s externalizing behavior. They found that programs in which parents were trained in problem solving showed smaller effects on positive parenting behavior and acquisition of parenting skills than programs in which parents were not trained in problem solving behavior. Possibly, parental personal and problem-solving skills are less directly related to child maltreatment than, for example, skills regarding parenting or the parent-child relationship. As parent training programs are often limited in their duration, programs with a main focus on the improvement of parental personal skills and problem-solving skills may not or to a smaller extent address parenting issues, or issues related to the parent-child relationship, which are more directly associated with child maltreatment. This could explain the negative moderating effects of these program components.

A negative moderating effect was also found for the component regarding the encouragement of the child’s pro-social behavior (or discouragement of anti-social behavior). This means that parent training programs promoting parents to stimulate more pro-social or less anti-social behavior of their child have smaller effects than programs not stimulating this child behavior. This result also corresponds with the findings of Kaminski et al. [33], as they found that programs promoting children’s (pro-) social skills showed smaller effects on positive parenting behavior and skills than programs without this focus. Previous research showed that anti-social behavior of children and child maltreatment are strongly associated [61,62,63]. However, in this research, the child’s anti-social behavior is more often considered as an important outcome than a cause for child maltreatment. Therefore, targeting the child’s anti-social behavior in parent training programs might not necessarily increase their effectiveness. Furthermore, when too much time and effort is spent on the child’s anti-social behavior, rather than on spending time and effort on parenting related issues, the program effectiveness may actually decrease, as these issues are more predictive of child maltreatment than factors related to the characteristics of the child [15,16,17]. This could explain the negative moderating effect of this program component. 

Finally, we found a negative moderating effect of the delivery technique practice and rehearsal. This finding suggests that parent training programs, in which certain skills are practiced in the program sessions by rehearsal and direct feedback of the trainer, are less effective than programs in which this technique is not applied. This is not in line with findings of Kaminski et al. [33], who reported that parental practice and rehearsal in training sessions was associated with better parent and child outcomes. However, the everyday life of high-risk or maltreating families could be very different from the setting in which new skills are practiced in the parent training programs. It is to be expected that multiple complex problems are present in these families, that cannot all be addressed in practice and rehearsal sessions. After all, from theory can be derived that child maltreatment is caused by a complex interaction between multiple risk factors, rather than by the presence of only a single risk factor [10,11]. It may thus be difficult to simulate everyday situations that high-risk or maltreating families have to deal with in practice and rehearsal sessions.

Although no specific contextual factors, structural elements or program components were found that significantly contributed to the overall effect of parent training programs, there are a number of common factors that all interventions need in order to be effective, regardless of the target group and the type of intervention. For example, programs should have a clear structure and a clear goal [64,65]. Furthermore, programs should be delivered as intended (i.e., according to the manual or protocol), as a higher level of program integrity is associated with larger significant effects of programs on various outcomes [66,67]. Finally, the relationship with the professional who carries out the program is a very important factor in parent training programs. Previous literature shows that a better quality of the parent-professional alliance is associated with larger improvements in child outcomes and parenting practices [68,69,70]. 

### 4.3. Limitations

Several limitations of the present meta-analysis should be discussed. First, the results of the current study do not permit conclusions about causality because of the non-experimental nature of this review. We investigated whether the presence or absence of individual program components or techniques were associated with the program effectiveness (expressed in an estimated overall effect) of parent training programs in reducing or preventing child maltreatment. Our findings indicate that none of the components or techniques significantly moderated the overall effect of parent training programs. However, it remains unclear whether the components or techniques were effective on itself. This should be investigated in future experimental research by assigning participants to either a group receiving an intervention with a certain component or a group receiving an intervention without this component. 

The second limitation is related to the outcomes that are measured in the included primary studies. Primary studies reported on outcome measures such as official reports, investigations, or a recurrence in child protection, as it was assumed that these measures were indicative of child maltreatment. However, previous literature suggested that a large proportion of maltreatment is not reported to the child protection authorities [71,72,73]. Therefore, these outcome measures may not be fully indicative of actual episodes of child abuse or neglect. Furthermore, we included studies reporting on measures of important indicators of child maltreatment, such as harsh parenting, corporal punishment, and parental aggression directed to the child, as previous literature points to a fine line between these terms and child maltreatment [74,75,76,77]. 

The third limitation is related to the sex of the parents who participated in the included primary studies. Most of these studies mainly recruited mothers, and some studies focused even solely on mothers in examining the effects of a parent training program. This is despite the fact that previous literature revealed that targeting fathers in parent training programs enhances child behavior and parent practices [78,79,80]. Therefore, parent training programs should seek to understand how to actively engage fathers and future research should examine treatment outcomes of programs involving fathers.

Finally, shortcomings regarding the coding of the program components and delivery techniques should be mentioned here. First, the information about the specific content of the parent training programs reported in the primary studies or the online factsheets and protocols was sometimes limited. Therefore, it was difficult to determine whether a program focused on a specific component or used a certain delivery technique or not. In addition, the way in which a parent training program was offered to participants of a study may differ from the implementation procedures that are described in the factsheets, manuals, and/or protocols that were found online. The scarcity of information on a program’s content may have caused incorrect codings of components and techniques. These shortcomings were noticed in the first coding round, in which some studies were double-coded (see also the Method section). It was decided that, in order to reduce subjectivity in the coding process, components were coded rather strictly. This means that a component or technique was only coded as present if the text clearly stated that the component or technique was part of the program content.

### 4.4. Implications for Clinical Practice and Future Research

Despite these limitations, our study provides important knowledge for clinical practice and suggestions for future research. Parent training programs have a significant effect on child maltreatment. Therefore, parent training programs should be taken into account by clinical professionals and policy makers in choosing an appropriate intervention to prevent or reduce child maltreatment, especially because most program components and delivery techniques used in these programs seem equally effective. However, improving parental personal skills, problem solving skills, and stimulating children’s prosocial behavior should not be the main focus of these programs, just as practicing new skills by rehearsal and giving direct feedback in program sessions. 

As our findings indicate that all program components and delivery techniques are about equally effective in preventing or reducing child maltreatment, it can be argued to combine components when treating high-risk or maltreating parents in parent training programs. The program components and techniques were only tested in bivariate models in this study. Therefore, no conclusions can be drawn about the interactions between components, and the effects of these interactions on child maltreatment. This should be examined in future research. Furthermore, it is very important that future intervention research gives a more specific and elaborate description of the intervention content, as this will greatly improve the quality of future component research.

## 5. Conclusions

The results of this meta-analysis showed a significant and small effect of parent training programs on child maltreatment. We found that the overall effect was moderated by several study characteristics (i.e., larger sample sizes, RCTs (versus quasi-experimental designs), and a larger study quality index yielded smaller effect sizes). No significant moderating effects were found for contextual factors and structural elements (i.e., duration, number of sessions, delivery location, and delivery setting). Notably, most of the program components and techniques that we investigated were not significantly associated with the effectiveness of parent training programs for preventing or reducing child maltreatment. Therefore, our findings indicate that most components are about equally effective and that none of the currently examined components or techniques are individually associated with the effectiveness of parent training programs.

## Figures and Tables

**Figure 1 ijerph-16-02404-f001:**
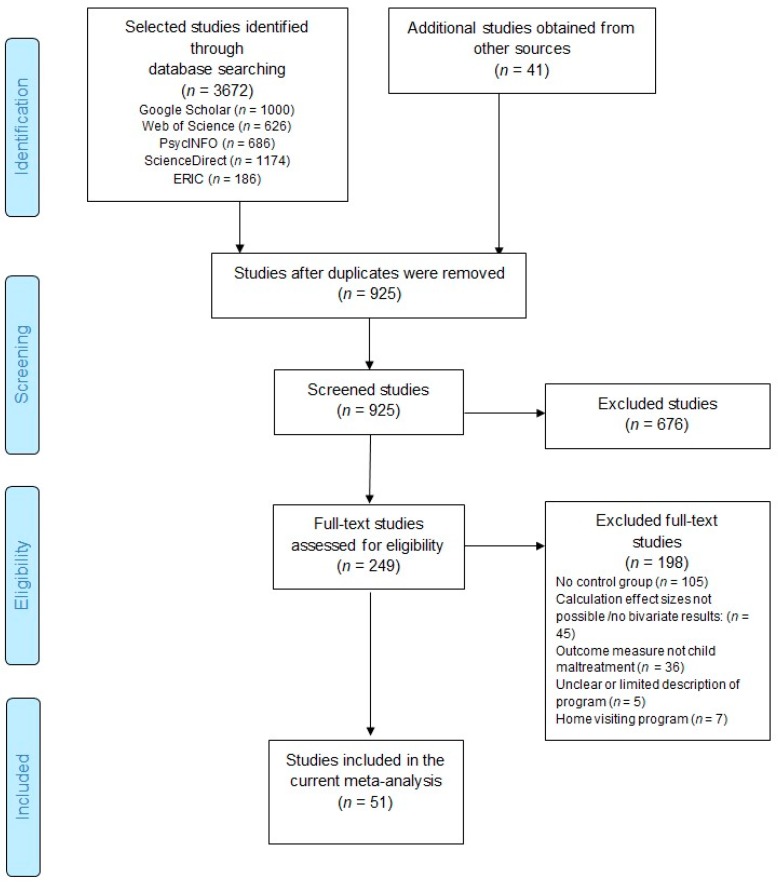
Flowchart of study selection procedure, according to the Preferred Reporting Items for Systematic Review and Meta-Analysis (PRISMA).

**Figure 2 ijerph-16-02404-f002:**
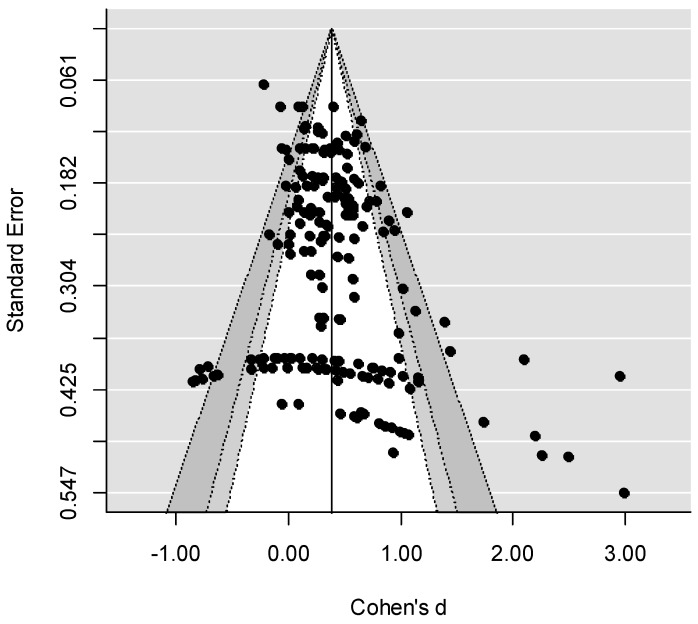
Funnel plot.

**Table 1 ijerph-16-02404-t001:** Descriptions of the coded program components and delivery techniques and their prevalence.

Program Components	%	Description
(1) Parent-child relationship in general	83.8	Improving parental skills regarding the parent-child relationship (including program components 2–5).
(2) Parent-child communication	64.9	Improving the communicative skills of parents in interaction with their child, and learning parents to interact in a positive way with their child.
(3) Affection, sensitivity, and/or responsivity	24.9	Improving affective behavior of parents towards their child, such as holding and cuddling their child, and responding sensitively to the child’s emotional and psychological needs.
(4) Quality time	14.1	Encouraging parents to spend quality time with the child, i.e., playing with the child, and doing fun activities together.
(5) Parent-child attachment	13.5	Stimulating a safe parent-child attachment.
(6) Disciplining skills in general	97.8	Improving parental skills regarding disciplining the child (including program components 7–12).
(7) Clear rules/consequences	50.3	Improving parental disciplinary communication skills, such as giving clear directions, setting limits and rules, and stating behavioral expectations and consequences.
(8) Time-out	58.9	Encouraging parents to use time-out as a disciplinary technique.
(9) Planned ignoring	62.7	Encouraging parents to ignore certain ‘bad’ or attention-seeking behaviors of their child as a disciplinary technique.
(10) Positive reinforcement	70.3	Encouraging parents to use positive reinforcement, such as praise and rewards, and to reinforce ‘good’ or prosocial behaviors of the child.
(11) Negative punishment	48.6	Encouraging parents to use negative consequences for ‘bad’ behavior of the child, such as taking away privileges, as a disciplinary technique.
(12) Alternatives for negative/physical discipline	70.3	Encouraging parents to use alternative parenting techniques for their negative parenting or physical discipline.
(13) Parental personal skills in general	95.1	All parental personal skills (including program components 14–20).
(14) Problem solving	64.9	Improving parental problem-solving skills.
(15) Stress management	23.2	Applying stress management strategies, such as meditation and other relaxation exercises, in order to reduce parental stress.
(16) Anger management	47.6	Applying anger/emotion management strategies, such as calming down, in order to reduce parental anger towards the child.
(17) Goal setting	20.0	Encouraging parents to select goals that are based on their own values, beliefs, and traditions.
(18) Cognitive skills	31.4	Improving cognitive (behavioral) skills of parents, including effective coping strategies.
(19) Listening skills	15.7	Stimulating parental attentive and active listening to their child.
(20) Being a role model	26.5	Encouraging parents to be a good role model for their child.
(21) Components regarding the stimulation of children’s skills	88.8	Encouraging parents to stimulate all skills of children (including program component 22–24).
(22) Pro-social/less anti-social behavior	64.9	Encouraging parents to stimulate pro-social behavior of children or to discourage anti-social behavior.
(23) Social skills	46.5	Stimulating the development of social skills of children, such as playing with others and cooperating.
(24) Cognitive/academic skills	23.2	Stimulating the development of cognitive/academic skills of children, such as language development and school success.
Other components		
(25) Supervision	13.5	Improving monitoring and supervising practices of parents.
(26) Consistency	22.7	Encouraging parents to react on certain child behaviors in a consistent manner.
(27) Calm, clear language, positive tone	14.1	Encouraging parents to stay calm, use clear language, and/or a positive tone when giving instructions to their child.
(28) Anticipating ‘high-risk’ situations	14.6	Encouraging parents to identify and anticipate on high-risk situations (i.e., situations in which there is a high risk of parenting problems or child abuse, such as during shopping), for example by setting up a prevention plan.
(29) Knowledge of child/development	42.7	Improving parental knowledge of their child’s developmental stages, and their child’s behavior and needs so that parents are able to provide developmentally appropriate physical care and to foster their child’s positive social-emotional development.
(30) Attitudes	15.7	Decreasing parental negative attitudes/attribution towards parenting, their child, or their child’s behavior, for example by using attributional retraining.
(31) Expectations	43.8	Improving realistic expectations/beliefs of parents regarding their child and/or parenting.
(32) Relation/collaboration parents	40.5	Improving the relationship and cooperation between parents, for example by stimulating that parents support each other if their child behaves problematic, and by giving and receiving constructive feedback. Also addressing marital problems or partner issues can be addressed.
(33) Parental competence/empowerment	37.8	Empowering parents and increasing their sense of self-esteem and competence regarding parenting.
(34) Social network	20.5	Helping parents to create a strong social network and to increase their involvement with the community, such as the school their child is attending.
Delivery techniques		
(1) Modelling	47.0	Giving live demonstrations of proper parenting behaviors or other forms of proper behavior.
(2) Role-playing	40.0	Practicing skills in program sessions by role-playing, either with the trainer or a peer (in a parenting group).
(3) Practice and rehearsal	69.7	Practicing skills (with a child) in the program sessions by rehearsal and direct feedback of the trainer.
(4) Video-feedback	0.5	Video-recording of parenting skills or parent-child interactions, so that the trainer has the opportunity to give feedback on video-recorded behavior and that parents can critique their own behavior.
(5) Homework assignments	70.8	Written, verbal, or behavioral assignments that are to be complete between sessions, including keeping a diary or practicing skills at home.
(6) (Group) discussion	67.0	Discussing parenting skills, either with an individual parent or in a group.
(7) CBT techniques for parents	11.9	Using cognitive behavioral therapy techniques (i.e., cognitive restructuring) or mindfulness techniques.
(8) Services for children	13.0	Having a child participate in a behavioral, social, cognitive, or social skills training separately from the parent.
(9) Additional services for parents	17.8	Providing additional services for parents which are not specifically aimed at improving parenting skills, such as offering social support and/or practical support, or referring parents for mental health or addiction problems.

Note. % = the percentage of effect sizes linked to the corresponding element or technique. In total, there were 185 effect sizes extracted from all included primary studies.

**Table 2 ijerph-16-02404-t002:** Overall effect for parent training programs on child maltreatment and sensitivity analysis.

Overall Effect	^#^ Studies	^#^ ES	Mean *d* (SE)	95% CI	Sig. Mean *d* (*p*)	% Var. at Level 1	Level 2 Variance	% Var. at Level 2	Level 3 Variance	% Var. at Level 3
Overall effect	50	185	0.416 (0.042) ***	(0.334, 0.498)	<0.0010 ***	33.3	0.053 ***	39.3	0.037 *	27.4
Overall effect without Kolko [53]	49	129	0.425 (0.047) ***	(0.331, 0.518)	<0.0010 ***	28.6	0.012 *	9.9	0.074 ***	61.5

Notes. ^#^ Studies = number of studies; ^#^ ES = number of effect sizes; Mean *d* = mean effect size (Cohen’s *d*); SE = standard error; CI = confidence interval; Sig. = significance; % Var. = percentage of distributed variance; level 1 variance = sampling variance; level 2 variance = variance within studies; level 3 variance = variance between studies.* *p* < 0.05; *** *p* < 0.001.

**Table 3 ijerph-16-02404-t003:** Results for the moderator analyses.

Moderator Variables	^#^ Studies	^#^ ES	Intercept/Mean *d* (95% CI)	β_1_(95% CI)	*F* (df1, df2) ᵃ	*p* ᵇ	Level 2 Variance	Level 3 Variance
**Overall Effect**	50	185	0.416 (0.334, 0.498) ***				0.232 ***	0.193 *
**A: Study characteristics**							
Publication year	50	185	0.449 (0.355, 0.544) ***	−0.009 (−0.021, 0.003)	1.968 (1, 183)	0.162	0.051 ***	0.037 **
***Sample characteristics***								
Type of sample					0.623 (2, 182)	0.537	0.051 ***	0.044 *
Risk group (RC)	32	89	0.419 (0.314, 0.525) ***					
General sample	6	9	0.543 (0.278, 0.808) ***	0.124 (−0.162, 0.409)				
Maltreating sample	13	87	0.365 (0.196, 0.534) ***	−0.054 (−0.250, 0.142)				
Sample size	50	185	0.448 (0.370, 0.525) ***	−0.001 (−0.001, −0.000) **	69.058 (1, 183)	0.003 **	0.052 ***	0.022 *
Age category child								
Unborn child/baby (≤2)					0.026 (1, 182)	0.872	0.052 ***	0.043 *
No (RC)	35	153	0.428 (0.326, 0.529) ***					
Yes	14	31	0.412 (0.249, 0.575) ***	−0.016 (−0.207, 0.176)				
Infant/toddler (2-5)					0.248 (1, 182)	0.619	0.051 ***	0.044 *
No (RC)	13	96	0.391 (0.235, 0.556) ***					
Yes	36	88	0.438 (0.334, 0.542) ***	0.047 (−0.140, 0.234)				
Primary school (6-12)					1.809 (1, 182)	0.180	0.051 ***	0.042 *
No (RC)	11	26	0.530 (0.351, 0.709) ***					
Yes	38	158	0.391 (0.294, 0.488) ***	−0.139 (−0.342, 0.065)				
High school (≥12)					2.616 (1, 182)	0.108	0.052 ***	0.035 *
No (RC)	41	108	0.452 (0.361, 0.542) ***					
Yes	8	76	0.282 (0.095, 0.469) ***	−0.170 (−0.378, 0.037)				
Age of child (average)	39	164	0.419 (0.303, 0.534) ***	−0.017 (−0.049, 0.015)	1.109 (1, 162)	0.294	0.056 ***	0.067 *
Age of the parent(s) (average)	46	126	0.420 (0.324, 0.516) ***	−0.016 (−0.033, 0.001) ^+^	3.621 (1, 124)	0.059 ^+^	0.012 *	0.072 ***
Percentage cultural minorities	29	136	0.401 (0.318, 0.484) ***	0.042 (−0.254, 0.337)	0.077 (1, 134)	0.781	0.087 ***	0.005
***Design characteristics***								
Research design					6.770 (2, 182)	0.001 **	0.047 ***	0.029
RCT (RC)	40	162	0.358 (0.274, 0.441) ***					
Quasi-experimental, matched	3	9	0.388 (0.094, 0.683) *	0.031 (−0.276, 0.337)				
Quasi-experimental, not matched	7	14	0.805 (0.580, 1.031) ***	0.448 (0.207, 0.688) ***				
Intent-to-treat design					2.320 (1, 183)	0.129	0.050 ***	0.040 **
No (RC)	23	99	0.495 (0.364, 0.626) ***					
Yes	27	86	0.364 (0.257, 0.472) ***	−0.131 (−0.300, 0.039)				
Min. 12 month follow-up					0.542 (1, 183)	0.463	0.053 ***	0.038 *
No (RC)	38	91	0.396 (0.297, 0.495) ***					
Yes	12	94	0.463 (0.313, 0.614) ***	0.067 (−0.113, 0.247)				
Study quality index (numerical score combining previous three variables)	50	185	0.396 (0.313, 0.478) ***	−0.112 (−0.195, −0.030) **	7.264 (1, 183)	0.008 **	0.047 ***	0.037 *
Control group					0.683 (3, 176)	0.564	0.054 ***	0.046 *
Treatment as usual (TAU; RC)	28	120	0.424 (0.299, 0.549) ***					
No treatment	4	11	0.493 (0.219, 0.768) ***	0.070 (−0.232, 0.371)				
Waiting list	12	35	0.318 (0.144, 0.492) ***	−0.106 (−0.321, 0.108)				
Other	7	11	0.507 (0.262, 0.752) ***	0.083 (−0.192, 0.358)				
***Outcome characteristics***								
Assessment type					2.062 (3, 181)	0.107	0.049 ***	0.036 ^+^
Self-report parents (RC)	44	137	0.391 (0.305, 0.477) ***					
Official reports	9	15	0.646 (0.428, 0.865) ***	0.255 (0.030, 0.481) *				
Observations	3	8	0.462 (0.143, 0.782) **	0.071 (−0.256, 0.398)				
Child-report	4	25	0.316 (0.124, 0.508) **	−0.075 (−0.261, 0.110)				
Follow-up duration (in months)	27	96	0.446 (0.345, 0.547) ***	0.002 (−0.006, 0.009)	0.195 (1, 94)	0.660	0.062 **	0.016
**B: Contextual factors**								
General aim of the program					0.558 (1, 183)	0.456	0.053 ***	0.038 *
Prevention (RC)	38	98	0.434 (0.339, 0.529) ***					
Reduction	13	87	0.364 (0.200, 0.527) ***	−0.071 (−0.257, 0.116)				
Delivery location								
Home/ambulant					0.267 (1, 183)	0.606	0.052 ***	0.041 *
No (RC)	30	73	0.436 (0.325, 0.547) ***					
Yes	21	112	0.393 (0.267, 0.519) ***	−0.043 (−0.209, 0.122)				
Treatment center					0.478 (1, 183)	0.490	0.052 ***	0.041 *
No (RC)	13	44	0.468 (0.302, 0.633) ***					
Yes	37	141	0.400 (0.302, 0.499) ***	−0.067 (−0.260, 0.125)				
By telephone					0.711 (1, 183)	0.400	0.048 ***	0.045 *
No (RC)	47	177	0.411 (0.324, 0.498) ***					
Yes	4	8	0.531 (0.254, 0.808) ***	0.120 (−0.160, 0.400)				
Online					0.214 (1, 183)	0.644	0.051 ***	0.043 *
No (RC)	47	165	0.425 (0.335, 0.515) ***					
Yes	3	20	0.359 (0.090, 0.627) **	−0.066 (−0.349, 0.217)				
Other					1.203 (1, 183)	0.274	0.029 ***	0.042 *
No (RC)	46	170	0.403 (0.315, 0.491) ***					
Yes	6	15	0.550 (0.297, 0.7803) ***	0.147 (−0.117, 0.411)				
Delivery setting								
Only parent(s)					0.484 (1, 181)	0.487	0.054 ***	0.042 *
No (RC)	36	134	0.403 (0.302, 0.505) ***					
Yes	12	49	0.471 (0.309, 0.632) ***	0.067 (−0.124, 0.258)				
Both parent(s) and child					1.263 (1, 181)	0.263	0.053 ***	0.041 *
No (RC)	32	112	0.451 (0.351, 0.552) ***					
Yes	17	71	0.368 (0.240, 0.495) ***	−0.084 (−0.231, 0.063)				
Parent group (without children)					0.227 (1, 181)	0.634	0.052 ***	0.045 *
No (RC)	20	94	0.404 (0.284, 0.523) ***					
Yes	29	89	0.439 (0.331, 0.547) ***	0.035 (−0.110, 0.180)				
**C: Structural elements**								
Duration					0.235 (2, 182)	0.790	0.050 ***	0.046 *
13–24 weeks (RC)	10	85	0.364 (0.182, 0.546) ***					
0–12 weeks	30	70	0.432 (0.318, 0.546) ***	0.068 (−0.147, 0.283)				
>24 weeks	11	30	0.444 (0.259, 0.629) ***	0.080 (−0.180, 0.339)				
Minimum duration (in weeks)	22	65	0.450 (0.319, 0.582) ***	0.004 (−0.011, 0.018)	0.247 (1, 63)	0.621	0.006	0.056 **
Maximum duration (in weeks)	47	181	0.409 (0.330, 0.488) ***	0.000 (−0.004, 0.005)	0.035 (1, 179)	0.851	0.065 ***	0.024 ^+^
Average number of sessions	33	88	0.346 (0.260, 0.432) ***	0.003 (−0.004, 0.010)	0.671 (1, 86)	0.415	0.015 **	0.030 *
Interval sessions					1.059 (3, 154)	0.368	0.110 ***	0.000
Weekly (RC)	28	116	0.363 (0.282, 0.445) ***					
Multiple sessions a week	8	31	0.511 (0.358, 0.664) ***	0.148 (−0.025, 0.322) ^+^				
Every other week/monthly	3	6	0.489 (0.163, 0.816) **	0.126 (−0.210, 0.463)				
Ascending/descending intensity	2	5	0.371 (−0.004, 0.746) ^+^	0.008 (−0.376, 0.392)				
**D: Program components**								
Parent-child relationship in general					2.371 (1, 183)	0.125	0.052 ***	0.037 **
No (RC)	3	30	0.583 (0.354, 0.813) ***					
Yes	49	155	0.410 (0.327, 0.482) ***	−0.173 (−0.395, 0.049)				
Parent-child communication					0.002 (1, 183)	0.969	0.053 ***	0.040 *
No (RC)	19	65	0.419 (0.294, 0.544) ***					
Yes	32	120	0.416 (0.318, 0.514) ***	−0.003 (−0.148, 0.142)				
Affection, sensitivity, and/or responsivity					1.417 (1, 183)	0.235	0.053 ***	0.037 *
No (RC)	33	136	0.453 (0.351, 0.556) ***					
Yes	18	46	0.350 (0.213, 0.487) ***	−0.103 (−0.274, 0.068)				
Quality time					3.168 (1, 183)	0.077 ^+^	0.052 ***	0.034 *
No (RC)	41	159	0.382 (0.294, 0.470) ***					
Yes	10	26	0.574 (0.380, 0.769) ***	0.193 (−0.021, 0.406) ^+^				
Parent-child attachment					1.016 (1, 183)	0.315	0.053 ***	0.038 *
No (RC)	38	126	0.439 (0.345, 0.533) ***					
Yes	13	25	0.338 (0.164, 0.513) ***	−0.101 (−0.299, 0.097)				
Disciplining skills					0.005 (1, 183)	0.946	0.053 ***	0.041 *
No (RC)	2	4	0.432 (0.004, 0.861) *					
Yes	48	181	0.417 (0.331, 0.503) ***	−0.015 (−0.452, 0.422)				
Clear rules/consequences				0.006 (1, 183)	0.940	0.052 ***	0.041 *	
No (RC)	17	92	0.422 (0.279, 0.564) ***					
Yes	34	93	0.415 (0.311, 0.520) ***	−0.007 (−0.183, 0.170)				
Time-out					1.054 (1, 183)	0.306	0.051 ***	0.043 *
No (RC)	22	76	0.377 (0.260, 0.493) ***					
Yes	30	109	0.450 (0.345, 0.556) ***	0.074 (−0.068, 0.216)				
Planned ignoring					0.148 (1, 183)	0.701	0.051 ***	0.042 *
No (RC)	21	69	0.401 (0.280, 0.522) ***					
Yes	30	116	0.429 (0.327, 0.530) ***	0.028 (−0.116, 0.171)				
Positive reinforcement					0.556 (1, 183)	0.457	0.051 ***	0.043 *
No (RC)	13	55	0.374 (0.231, 0.518) ***					
Yes	39	130	0.433 (0.339, 0.527) ***	0.059 (−0.096, 0.214)				
Negative punishment					0.025 (1, 183)	0.875	0.054 ***	0.038 *
No (RC)	29	95	0.422 (0.318, 0.526) ***					
Yes	22	90	0.410 (0.297, 0.524) ***	−0.011 (−0.151, 0.129)				
Alternatives for negative/physical discipline					1.588 (1, 183)	0.209	0.052 ***	0.040 *
No (RC)	20	55	0.485 (0.350, 0.620) ***					
Yes	31	130	0.375 (0.268, 0.482) ***	−0.110 (−0.281, 0.062)				
Personal skills of parents					10.520 (1, 183)	0.001 **	0.056 ***	0.022 ^+^
No (RC)	6	9	0.816 (0.558, 1.075) ***					
Yes	45	176	0.373 (0.297, 0.450) ***	−0.443 (−0.713, 0.174) **				
Problem solving					4.195 (1, 183)	0.042 *	0.047 ***	0.042 **
No (RC)	22	65	0.512 (0.388, 0.637) ***					
Yes	29	120	0.363 (0.265, 0.462) ***	−0.149 (−0.293, −0.005) *				
Stress management					1.431 (1, 183)	0.233	0.053 ***	0.037 *
No (RC)	34	142	0.451 (0.351, 0.551) ***					
Yes	17	43	0.345 (0.203, 0.488) ***	−0.106 (−0.280, 0.069)				
Anger management					2.631 (1, 183)	0.107	0.056 ***	0.027 *
No (RC)	27	97	0.464 (0.364, 0.564) ***					
Yes	25	88	0.355 (0.273, 0.500) ***	−0.109 (−0.241, 0.024)				
Goal setting					0.353 (1, 183)	0.553	0.052 ***	0.040 *
No (RC)	40	148	0.432 (0.335, 0.528) ***					
Yes	11	37	0.374 (0.206, 0.541) ***	−0.058 (−0.251, 0.125)				
Cognitive skills					0.423 (1, 183)	0.516	0.052 ***	0.042 *
No (RC)	39	127	0.405 (0.312, 0.498) ***					
Yes	14	58	0.456 (0.313, 0.598) ***	0.051 (−0.103, 0.205)				
Listening skills					0.791 (1, 183)	0.375	0.053 ***	0.039 *
No (RC)	39	156	0.437 (0.343, 0.530) ***					
Yes	11	29	0.344 (0.162, 0.526) ***	−0.092 (−0.297, 0.113)				
Being a role model					0.667 (1, 183)	0.415	0.054 ***	0.040 *
No (RC)	33	136	0.444 (0.338, 0.550) ***					
Yes	17	49	0.372 (0.234, 0.510) ***	−0.072 (−0.246, 0.102)				
Skills of children					2.232 (1, 183)	0.137	0.051 ***	0.041 *
No (RC)	40	21	0.547 (0.356, 0.738) ***					
Yes	11	164	0.386 (0.292, 0.480) ***	−0.161 (−0.374, 0.052)				
Pro-social/less anti-social behavior					5.134 (1, 183)	0.025 *	0.049 ***	0.039 **
No (RC)	18	65	0.527 (0.400, 0.655) ***					
Yes	33	120	0.361 (0.266, 0.457) ***	−0.166 (−0.311, 0.021) *				
Social skills					0.015 (1, 183)	0.903	0.053 ***	0.040 *
No (RC)	30	99	0.413 (0.309, 0.518) ***					
Yes	21	86	0.422 (0.307, 0.537) ***	0.009 (−0.142, 0.150)				
Cognitive/academic skills					0.759 (1, 183)	0.759	0.051 ***	0.041 *
No (RC)	34	142	0.443 (0.341, 0.545) ***					
Yes	16	43	0.364 (0.216, 0.511) ***	−0.079 (−0.258, 0.100)				
Other components								
Supervision					1.107 (1, 183)	0.294	0.048 ***	0.046 *
No (RC)	39	160	0.442 (0.346, 0.539) ***					
Yes	11	25	0.329 (0.141, 0.518) ***	−0.113 (−0.324, 0.099)				
Consistency					2.424 (1, 183)	0.121	0.051 ***	0.036 *
No (RC)	39	143	0.459 (0.361, 0.557) ***					
Yes	11	42	0.321 (0.176, 0.465) ***	−0.138 (−0.312, 0.037)				
Calm, clear language, positive tone					1.197 (1, 183)	0.275	0.052 ***	0.040 *
No (RC)	42	159	0.439 (0.336, 0.531) ***					
Yes	8	26	0.317 (−0.098, 0.340) **	−0.121 (−0.340, 0.098)				
Anticipating ‘high-risk’ situation					0.008 (1, 183)	0.927	0.052 ***	0.042 *
No (RC)	43	158	0.420 (0.327, 0.513) ***					
Yes	7	27	0.409 (0.206, 0.613) ***	−0.010 (−0.234, 0.213)				
Knowledge of child/development					0.431 (1, 183)	0.512	0.052 ***	0.042 *
No (RC)	16	106	0.458 (0.312, 0.603) ***					
Yes	35	79	0.398 (0.294, 0.502) ***	−0.060 (−0.238, 0.119)				
Attitudes					0.966 (1, 183)	0.327	0.052 ***	0.041 *
No (RC)	38	156	0.441 (0.345, 0.538) ***					
Yes	12	29	0.343 (0.171, 0.512) ***	−0.098 (−0.296, 0.099)				
Expectations					0.652 (1, 183)	0.420	0.051 ***	0.041 *
No (RC)	33	104	0.394 (0.292, 0.496) ***					
Yes	19	81	0.452 (0.333, 0.572) ***	0.058 (−0.084, 0.200)				
Relation/collaboration parents					1.071 (1, 183)	0.302	0.047 ***	0.046 *
No (RC)	38	110	0.443 (0.345, 0.540) ***					
Yes	15	75	0.366 (0.233, 0.498) ***	−0.077 (−0.224, 0.070)				
Parental competence/empowerment					1.313 (1, 183)	0.253	0.051 ***	0.040 *
No (RC)	28	115	0.462 (0.348, 0.577) ***					
Yes	23	70	0.366 (0.245, 0.487) ***	−0.097 (−0.263, 0.070)				
Social network					0.089 (1, 183)	0.766	0.053 ***	0.039 *
No (RC)	36	147	0.409 (0.310, 0.508) ***					
Yes	15	38	0.437 (0.283, 0.590) ***	0.028 (−0.155, 0.210)				
**E: Delivery techniques**								
Modelling					3.545 (1, 183)	0.061 ^+^	0.050 ***	0.036 *
No (RC)	18	98	0.313 (0.179, 0.477) ***					
Yes	33	87	0.474 (0.372, 0.575) ***	0.160 (−0.008, 0.328) ^+^				
Role-playing					0.185 (1, 183)	0.667	0.052 ***	0.041 *
No (RC)	22	111	0.438 (0.313, 0.563) ***					
Yes	28	74	0.401 (0.287, 0.551) ***	−0.037 (−0.206, 0.132)				
Practice and rehearsal					5.485 (1, 183)	0.020*	0.054 ***	0.028 ^+^
No (RC)	23	56	0.512 (0.398, 0.627) ***					
Yes	27	128	0.329 (0.225, 0.433) ***	−0.184 (−0.338, −0.029) *				
Video-feedback					1.358 (1, 183)	0.245	0.053 ***	0.035 *
No (RC)	49	184	0.412 (0.330, 0.494) ***					
Yes	1	1	0.984 (0.019, 1.949) *	0.572 (−0.397, 1.541)				
Homework assignments					0.000 (1, 183)	0.988	0.053 ***	0.039 *
No (RC)	17	54	0.418 (0.281, 0.554) ***					
Yes	34	131	0.417 (0.323, 0.510) ***	−0.001 (−0.152, 0.149)				
(Group)discussion					0.968 (1, 183)	0.327	0.048 ***	0.045 *
No (RC)	18	61	0.366 (0.230, 0.502) ***					
Yes	34	124	0.448 (0.344, 0.553) ***	0.082 (−0.083, 0.247)				
CBT techniques for parents					0.043 (1, 183)	0.836	0.053 ***	0.039 *
No (RC)	46	163	0.414 (0.327, 0.502) ***					
Yes	5	22	0.442 (0.191, 0.693) ***	0.028 (−0.236, 0.291)				
Services for children					0.206 (1, 183)	0.651	0.053 ***	0.040 *
No (RC)	44	161	0.425 (0.335, 0.516) ***					
Yes	6	24	0.372 (0.158, 0.586) ***	−0.053 (−0.286. 0.179)				
Additional services for parents					0.176 (1, 183)	0.675	0.051 ***	0.044 *
No (RC)	40	152	0.428 (0.332, 0.524) ***					
Yes	11	33	0.386 (0.209, 0.562) ***	−0.042 (−0.242, 0.157)				

Note. # Studies = number of studies; # ES = number of effect sizes; mean *d* = mean effect size Cohen’s *d*; CI = confidence interval; β_1_ = estimated regression coefficient; df = degrees of freedom; Level 2 variance = variance of effect sizes within studies; Level 3 variance = variance between studies. ᵃ Omnibus test of al regression coefficients of the model. ᵇ *p*-value of the omnibus test. ^+^
*p* < 0.1; * *p* < 0.05; ** *p* < 0.01; *** *p* < 0.001.

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
