# Peer review of "The Effectiveness of Parent Training Programs for Child Maltreatment and Their Components: A Meta-Analysis"

_ijerph, 2019, doi:10.3390/ijerph16132404_

Round 1
Reviewer 1 Report
Dear authors,
thank you very much for your interesting meta-analysis. Due to the high quality of your work I have only one question/remark: Could you please explain more in detail what it means that – following your analyis – there are no "moderating effects [...] for contextual factors and structural elements"? Please focus on the possible reasons and the consequences.
Furthermore, it would be interesting to know what framing conditions parenting programs need to be effective.
Best wishes.
Author Response
Response to Reviewer 1
Point 1: thank you very much for your interesting meta-analysis. Due to the high quality of your work I have only one question/remark: Could you please explain more in detail what it means that – following your analyis – there are no "moderating effects [...] for contextual factors and structural elements"? Please focus on the possible reasons and the consequences.
Response point 1: We agree with this reviewer that we could elaborate more on the findings regarding the contextual factors and structural elements. Therefore, we have added the following text to the Discussion: “Therefore, it is not expected that programs that differ on these characteristics will also differ in their effectiveness. This is not in line with the findings of Van der Put and colleagues [9] who found moderating effects for several structural element of child maltreatment interventions. However, they focused on a wide variety of child maltreatment interventions, including multisystemic interventions and home visiting programs, which are all very different in nature. The structural elements and contextual factors of the included parent training programs in this meta-analysis are more similar, which may explain why these variables didn’t significantly moderate the overall effect. Furthermore, our findings might be explained by the discrepancy between information that was used to code these characteristics (derived from factsheets, protocols and manuals) and the way the programs were actually performed within the research groups of the included studies. Possibly, some of the programs were performed more flexible instead of strictly according to the protocol. This makes it is difficult to categorize these programs with regard to their contextual factors and structural elements, as in reality these characteristics might be different. This should be taken into account when interpreting these results. However, there might be other structural and contextual factors, not investigated in the current study, that are related to the effectiveness of parent training programs” (p. 21).
Point 2: Furthermore, it would be interesting to know what framing conditions parenting programs need to be effective.
Response point2: Furthermore, we now added the following paragraph to the Discussion about which general framing conditions parent training programs need in order to be effective: “Although no specific contextual factors, structural elements or program components were found that significantly contributed to the overall effect of parent training programs, there are a number of common factors that all interventions need in order to be effective, regardless of the target group and the type of intervention. For example, programs should have a clear structure and a clear goal [64,65]. Furthermore, programs should be delivered as intended (i.e., according to the manual or protocol), as a higher level of program integrity is associated with larger significant effects of programs on various outcomes [66,67]. Finally, the relationship with the professional who carries out the program is a very important factor in parent training programs. Previous literature shows that a better quality of the parent-professional alliance is associated with larger improvements in child outcomes and parenting practices [68¬70]” (p. 22-23).
Reviewer 2 Report
This is an interesting piece of research, the quantatative aspects of which I am unqualified to review, that appears to be sound in methodology and conclusions. A major drawback is the omission of the sex of parents in the various studies reviewed, that is, in reality most of the existing works involve mainly mothers (and often mothers as the sole parent that is the subject of the intervention), and there are much less studies of fathers' invovlement in parenting programmes. We know that parenting interventions involving the mother and child or children alone are not as successful as those that seek to and achieve the involvement of both parents whether or not the father is resident in the family (see for example Panter-Brick et al, 2014). The addition of this consideration would add considerable authority to the paper.
Author Response
Response to Reviewer 2
Point 1: This is an interesting piece of research, the quantatative aspects of which I am unqualified to review, that appears to be sound in methodology and conclusions. A major drawback is the omission of the sex of parents in the various studies reviewed, that is, in reality most of the existing works involve mainly mothers (and often mothers as the sole parent that is the subject of the intervention), and there are much less studies of fathers' invovlement in parenting programmes. We know that parenting interventions involving the mother and child or children alone are not as successful as those that seek to and achieve the involvement of both parents whether or not the father is resident in the family (see for example Panter-Brick et al, 2014). The addition of this consideration would add considerable authority to the paper.
Response point 1: We agree that the omission of the sex of the parents in many of the included studies is an important limitation, and indeed, the included primary studies mainly examined intervention effects for mothers (and in 11 studies, solely mothers were recruited). Therefore, we have added the following text to the Limitations section: “The third limitation is related to the sex of the parents who participated in the included primary studies. Most of these studies mainly recruited mothers, and some studies focused even solely on mothers in examining the effects of a parent training program. This is despite the fact that previous literature revealed that targeting fathers in parent training programs enhances child behavior and parent practices [78¬80]. Therefore, parent training programs should seek to understand how to actively engage fathers and future research should examine treatment outcomes of programs involving fathers” (p. 23).
Reviewer 3 Report
Thank you for the opportunity to read this informative manuscript. I think the authors have made a great effort in explaining the background, and their study implies a new step towards the understanding the effectiveness of parent training programs for child maltreatment. In my opinion, the manuscript is well written, properly documented, justified and organized, results are generally well presented and conclusions are directly derived from the results. In my opinion the article merits publication.
My main concern refers to the explanation of the methods:
Major concerns:
1) What time frame was used to the studies selection? In the studies selection procedure the authors stated that studies were searched until July 2018 but they do not indicated if the establish a publication year to start the search. However, in the results section the authors stated that the studies were published between 1985 and 2018. Authors should include a rationale to explain the relevance of that time frame. Please, clarify.
2) In the screening phase, authors may indicate the criteria followed to exclude 676 studies. It would be good that authors include the criteria in the flowchart as they did in the eligibility phase.
3) I do not understand why the authors did not doubled-coded all the studies. Indeed, the internal validity of the study would have increased if the coding process had been conducted by a triangulation of the three authors. Please, justify.
Minor concerns:
In page 2, line 60 authors stated: “When looking at the results of some recent review studies…” The author should considered to deleted the word “recent” because some of the cited studies is from 2009 and when the time come no one will be recent.
Given that subsection 2.5 and 2.6 refers to statistical analysis I think it is appropriate to join both subsections and delete one of the headings.
First paragraph in page 12 must be deleted. I guess that paragraph is part of the template of the journal and it is not relevant for the study.
Author Response
Response to Reviewer 3
Thank you for the opportunity to read this informative manuscript. I think the authors have made a great effort in explaining the background, and their study implies a new step towards the understanding the effectiveness of parent training programs for child maltreatment. In my opinion, the manuscript is well written, properly documented, justified and organized, results are generally well presented and conclusions are directly derived from the results. In my opinion the article merits publication.
My main concern refers to the explanation of the methods:
Major concerns:
1. What time frame was used to the studies selection? In the studies selection procedure the authors stated that studies were searched until July 2018 but they do not indicated if the establish a publication year to start the search. However, in the results section the authors stated that the studies were published between 1985 and 2018. Authors should include a rationale to explain the relevance of that time frame. Please, clarify.
We did not apply a time frame to our search procedure. We searched until July 2018, meaning that all relevant studies (that met out inclusion criteria) published before July 2018 were included. We have now described this explicitly in the Methods section on page 4. The statement in the Results section about the publication year of studies was only meant to give a description of the included studies in this meta-analysis.
2. In the screening phase, authors may indicate the criteria followed to exclude 676 studies. It would be good that authors include the criteria in the flowchart as they did in the eligibility phase.
This flowchart was based on the PRISMA flow diagram and guidelines (see http://prismastatement.org/PRISMAStatement/FlowDiagram), which is used as a standard in systematic reviews and meta-analyses to depict the flow of information through the different phases (we have now mentioned this in the caption of Figure 1). In this flow diagram, it is not usual to provide the criteria for the studies that were excluded in the screening phase. These studies were excluded based on their title and/or abstract, because of their irrelevance to the current meta-analysis. We have now elaborated on this in the revised manuscript: “The searches resulted in 3,713 studies. After removing duplicates, 925 studies were screened based on their title and abstract. In the screening phase, 676 studies were excluded because of their irrelevance to the subject of this meta-analysis (e.g. studies examining other types of programs or other outcomes). Of the remaining 249 relevant studies the full text was evaluated. Finally, 51 studies met all inclusion criteria and were included in the current study” (p. 4-5). Furthermore, we changed the text in the ‘screening’ section of the flow-chart to: “Irrelevant studies excluded based on title and abstract” (p. 5).
3. I do not understand why the authors did not doubled-coded all the studies. Indeed, the internal validity of the study would have increased if the coding process had been conducted by a triangulation of the three authors. Please, justify.
We agree with this reviewer that it would be best to double-code all studies. However, given the available time and resources, we decided to code a sample of the included studies, as we found that the discrepancies in the double-coded study characteristics, the contextual factors, the structural elements, and effect sizes were the result of minor coding errors and differences in the reporting of the effect sizes between the coders (i.e., rounding differences). Furthermore, after double-coding the program components and delivery techniques for the sample of five double-coded studies, clear decisions were made on how to code these components and techniques for the remaining studies. Also, whenever the first author doubted about the presence of a certain component or technique, the other two authors were consulted (we have now elaborated this on page 9 of the manuscript). Therefore, we assume that our decision not to double-code all studies and effect sizes did not greatly affected our results. If this reviewer believes that double-coding all included studies is still required, we are off course willing to do that. However, this might be problematic with regard to the submission deadline as double-coding all studies is a very time-consuming process.
Minor concerns:
4. In page 2, line 60 authors stated: “When looking at the results of some recent review studies…” The author should considered to deleted the word “recent” because some of the cited studies is from 2009 and when the time come no one will be recent.
We agree, and we therefore deleted the word “recent” (p. 2; ln. 61).
5. Given that subsection 2.5 and 2.6 refers to statistical analysis I think it is appropriate to join both subsections and delete one of the headings.
We apologize for the double subsection heading. Subsection 2.6 should have been “Bias Assessment”. Therefore, we have now changed this heading (p. 10; ln. 284). However, if this reviewer still believes that it is necessary to join both subsections, we are off course willing to do that.
6. First paragraph in page 12 must be deleted. I guess that paragraph is part of the
template of the journal and it is not relevant for the study.
This paragraph has now been deleted.
Additional adjustments
• Page 20, lines 360-363: this paragraph was in the wrong place (in the text of the
pervious paragraph on line 359)
Round 2
Reviewer 2 Report
With note of expanded section on limitations now added, paper now ok for acceptance.